# Assessing food safety practices and foodborne illness risk factors in Brazilian households

**Gustavo Guimarães Fernandes Viana[1], Andréia Gonçalves Arruda[2]*,
Gabriel Augusto Marques Rossi[1]**

**1** Department of Veterinary Medicine, University of Vila Velha, Vila Velha, Espírito Santo, Brazil,
**2** Department of Veterinary Preventive Medicine, The Ohio State University, Columbus, Ohio, United States of America

* arruda.13@osu.edu

## Abstract

Foodborne illnesses represent a pressing public health issue, with Brazilian households accounting for over a third of reported outbreaks in the country. This study aimed to investigate food handling practices in Brazilian homes, evaluating their influence on the self-reported occurrence of foodborne illnesses. A total of 1,043 respondents participated in the study. Data was collected through an online questionnaire that captured their demographic data, feeding hygiene behaviors, food storage and preparation habits. Cluster analysis identified four groups based on adherence to food safety practices. Key findings revealed significant deficiencies: only 3.07% utilize thermometer to check meat doneness, which prevents the ingestion of viable pathogens, 64.90% of participants use the same cutting board for raw meat and vegetables and only 32.70% reported washing packages before storing them in the refrigerator. Risk factor analysis highlighted that older adults and males reported fewer foodborne illness, while cluster that adhered less to proper food manipulation exhibited higher illness rates. Our findings emphasize the urgent need for targeted educational campaigns to address critical gaps, such as discouraging meat washing and promoting safe storage practices. This study underscores the importance of innovative interventions to reduce household foodborne illnesses, contributing to improved public health outcomes.

## Introduction

Foodborne illness remain one of the most significant public health issues globally, accounting for a large proportion of preventable deaths and economic costs [1]. Between 2013 and 2022, Brazil reported 6,523 outbreaks of foodborne illnesses, 35.1% of which occurred in households [2]. However, these figures are likely underreported, as symptoms are often mild, and many affected individuals do not seek medical attention. Furthermore, the diversity and nonspecificity of symptoms complicate the detection and notification of foodborne illness cases [3].

**Data availability statement:** All relevant data are within the paper and its Supporting Information files.

**Funding:** Fundação de Amparo à Pesquisa e Inovação do Espírito Santo (FAPES) (Grant 551/2023 P 2023-RH7P2).

**Competing interests:** The authors have declared that no competing interests exist.

Most food consumed in the world is prepared at home, providing plentiful opportunities for mishandling and inadequate preservation, which in turn heightens the risk of household foodborne illness. Unlike commercial establishments, domestic kitchens serve multifunctional purposes, extending beyond food preparation and storage, such as acting spaces for working, socializing, presence of pets and presence of highly contaminated items such as mobile phones and purses. This multifunctionality increases the frequency of cross-contamination events [4].

The occurrence of household foodborne illness outbreaks highlights a critical issue: the lack of sanitary education and knowledge about proper food preparation and storage among the general population [5]. Furthermore, household outbreaks often receive less attention because they affect fewer individuals compared to outbreaks in restaurants and food service establishments. Consequently, this contributes to the limited focus of educational campaigns and targeted training initiatives [6].

While there is a growing body of research on food safety in commercial and industrial settings, studies focusing specifically on household practices remain scarce, especially in Brazil. Key gaps include understanding the socioeconomic and regional factors that influence behaviors related to food handling and preparation.

Existing literature show conflicting results when investigating the impact of demographical factors on food handling practices, moreover, most studies involve modest sample sizes [7–11]. Given that population habits are constantly evolving, particularly after the COVID-19 pandemic, it is crucial to obtain up-to-date knowledge about food handling practices. The pandemic has significantly altered how individuals approach food safety and foodborne illness prevention. Heightened awareness of contamination pathways has led to increased adherence to hand hygiene practices and a reduction in common food handling errors [12,13]. Thus, studies that explore these behaviors in the post-pandemic period are essential to understanding population-level changes, especially when comparing pre- and post-pandemic findings.

Although studies investigating food safety practices at home often employ questionnaires as their primary data collection method, there is no consistent methodological framework across the existing literature. Most prior research relies on simple statistical tests, such as chi-square or t-tests, to analyze their findings [14–18]. In contrast, this study integrates cluster analysis with risk factor analysis, allowing for the identification of behavioral patterns and their specific associations with the likelihood of foodborne illness. This innovative combination enhances our ability to uncover nuanced relationships in the data, offering unique insights into household food safety practices.

This study aimed to describe the knowledge and implementation of appropriate food handling practices in Brazilian households; and investigate the effect of such behaviors (risk factors) in self-reported household cases of foodborne illness. This will allow for the identification of critical points of failure and development of strategies for health education and communication aimed at preventing foodborne illness and promoting food safety in Brazilian households.

## Materials and methods

The study was approved by the Vila Velha University Research Ethics Committee (protocol code 69770523.9.0000.5064 and date of approval 26 June 2023) and informed consent was obtained from all subjects involved in the study. No minors were included in the study.

### The questionnaire

A digital questionnaire was administered using Google Forms. Participation was voluntary, with the link shared through various social media platforms such as Facebook, Instagram, and WhatsApp. Additionally, a flyer containing a QR code linked to the form was distributed in multiple locations at Vila Velha University, Vila Velha, state of Espírito Santo, Brazil. As such, an estimation of number of people recruited for participation was unavailable. The questionnaire was available for responses from September 4, 2023 to April 26, 2024. Of the 1,047 people that opened the survey link, three did not agree to the study's informed consent form and one participant was under the age of 18; all of which were excluded from further analysis. In total, 1,043 respondents provided written consent for the study, which was approved by the Vila Velha University Research Ethics Committee (CEP-UVV, protocol 69770523.9.0000.5064).

The questionnaire was designed to capture a comprehensive range of variables related to household food safety practices and respondent profiles. The selection of the questions was based on a review of existing literature on the development of questionnaires to evaluate food safety at home [19–21] as well as guidelines provided by global health organizations [22,23]. Additionally, questions exploring demographic and socioeconomic characteristics were included to assess their influence on food handling practices and self-reported foodborne illness.

The questionnaire contained 41 total questions divided in 9 Blocks; Block 0: Demographic Information (5 questions); Block 1: Personal Hygiene and Food Hygiene (6 questions); Block 2: Cleaning of Utensils and Surfaces (6 questions); Block 3: Temperature Stability (4 questions); Block 4: Refrigerator Cleaning, Usage, and Maintenance (4 questions); Block 5: Food Consumption (6 questions); Block 6: Presence and Usage of Kitchen Utensils (5 questions); Block 7: Consumption of High-Risk Foods (3 questions) and Block 8: Occurrence of Foodborne Illness in the Household (1 question). The full questionnaire is available in Supplementary Material Table S1.

### Statistical analysis

Statistical analyses were performed using Stata 14.4. Basic descriptive analyses (mean, standard deviation, range) were initially conducted; and categories within variables that had small sample sizes (e.g., variable 'education level') were appropriately combined for analysis. This was followed by cluster analysis, which was performed with the objective of grouping respondents based on similar answers for questionnaire Blocks 1–7 (excluding demographic questions and occurrence of foodborne illness question). The complete-linkage cluster method was used with a continuous dissimilarity measure and based on L2 or Euclidean distance. The number of clusters was validated by visualization of cluster dendrograms and by applying the Duda-Hart stopping rule [24]. Four distinct clusters were identified and subsequently used in risk factor analysis.

Risk factor analysis aimed to investigate the effect of cluster membership (main exposure of interest) on the self-reported occurrence of cases of foodborne illness in the household (dependent variable). Demographic-related variables namely participant region, education level, income, age and sex were included as independent variables. Model building followed the following steps: first, univariable logistic regression models were constructed for each predictor separately, with a $P < 0.2$ used as a cut-off. Next, Spearman correlation coefficients were used to check multicollinearity among variables, with a cutoff of 0.80. Final logistic multivariable models were built using a backward stepwise approach, and confounders were assessed constructing a causal diagram built a priori, and statistical assessment which included any change in other variable's coefficients larger than 20%. Statistical significance was declared at $P < 0.05$.

## Results

The demographical profile of study respondents is described in Table 1, for all participants and by cluster.

Among the most significant deficiencies in adopting proper food handling practices among respondents, (Table S1, Supplementary material), several findings stood out. Over half of the participants (52.73%) reported washing vegetables with only water, while 471 respondents (45.16%) stored food indiscriminately within the refrigerator, regardless of the type of food. Additionally, 702 respondents (67.30%) indicated they do not wash products before storing them in the refrigerator. The use of thermometers to verify meat doneness was particularly uncommon, with only 3.07% of participants employing this method. Furthermore, 677 respondents (64.90%) reported using the same cutting board for raw meat and vegetables.

Overall, the proportional distribution of demographic variables, including sex, income, and education level, appeared numerically similar consistent across clusters and aligned closely with the total respondent demographics. The only notable deviation was that Cluster 1 included a higher proportion of older respondents compared to the other clusters (Table 1).

Cluster membership appeared to reflect different reported attitudes in regard to knowledge and implementation of appropriate food handling practices in Brazilian households. Cluster 1 consisted of 136 individuals (13.03% of the total) who mostly adhered to good food handling practices. Respondents in this cluster reported overall higher frequencies of cleaning utensils, hands, and equipment. Cluster 1 members also reported the lowest consumption of undercooked meat (37.50% vs 43.18–53.93), raw meat (43.38% vs 57.95–64.22) and soft yolk egg (33.82% vs 38.64–45.04). However, they exhibited notable shortcomings compared to other clusters in regards to common use of trash bins on kitchen countertops (58.09%), mixing of chemical cleaning agents or disinfectants for kitchen sanitation (55.15%) and frequent washing of raw meats of all types before cooking (100%). These behaviors suggest that, while respondents in cluster 1 recognize the importance of hygiene in food handling and its role in contamination prevention, they lacked awareness of the risks associated with cross-contamination stemming from raw meat washing.

Cluster 2 represented an "intermediate adherence to proper food handling practices" group, with 88 respondents (8.43%) who neither excelled nor performed poorly in adopting good food handling practices. This cluster demonstrated moderate adherence to hygiene practices, for example, the vast majority of cluster 2 participants properly stored leftover food in refrigerators (95.46%). Conversely, in areas where overall adherence was low in this study, such as thermometer usage for assessing meat doneness, cluster 2 performed similarly poorly (2.27%). Of note, this cluster also displayed a high prevalence of raw meat washing (100%), indicating limited awareness of contamination risks despite moderate knowledge of other good practices.

Cluster 3 encompassed a large number of participants, 464 (44.44%), and had the weakest adherence to good food handling practices. This group exhibited the lowest cleaning frequencies in multiple questions, and were close to the highest consumption of undercooked meat (51.72% vs 37.50–53.93), the highest consumption of raw meat (64.22% vs 43.36–62.92), and the greatest tendency to consume expired products (40.95% vs 14.71–27.27). Unlike Cluster 1, where meat washing was a common but isolated issue, Cluster 3 rarely washed meats (0.86%), likely due to their overall lack of cleaning habits. Notably, cluster 3 members primarily relied on tap water for preparing broths and soups (51.51%), did not use separate cloths for utensils and hands (59.05%), and were the only group where most respondents consumed food past their secondary shelf life (61.85% vs 33.09–48.86).

Finally, cluster 4, comprising 356 participants (34.10%), rivaled Cluster 1 in terms of good food handling practices. However, several key distinctions emerged. Cluster 4 participants avoided raw meat washing (98.88% reported not washing meats), had lower trash bin usage on kitchen countertops (57.02% did not use it vs 41.91% of cluster 1), and stored eggs in more temperature-stable refrigerator compartments (72.47%) compared to cluster 1 (43.38%). On the other hand, they consumed more undercooked (53.93% vs 37.50% of cluster 1) and raw meat (62.92% vs 43.38% of cluster 1), and reported overall lower cleaning frequencies for utensils and surfaces.

**Table 1. Demographic profile of survey respondents by cluster in relation: Age group, sex, region, education level and monthly income.**

| Variable | Category | ALL (n = 1,043) | Cluster 1 (n = 136) | Cluster 2 (n = 88) | Cluster 3 (n = 464) | Cluster 4 (n = 355) |
|---|---|---|---|---|---|---|
| **Age** | 18–28 | 445 (42.66%) | 61 (44.85%) | 33 (37.50%) | 194 (41.81%) | 157 (44.25%) |
| | 29–39 | 347 (33.27%) | 15 (11.03%) | 31 (35.23%) | 183 (39.44%) | 118 (33.24%) |
| | 40–50 | 160 (15.34%) | 23 (16.91%) | 16 (18.18%) | 71 (15.30%) | 50 (14.08%) |
| | 51–61 | 60 (5.75%) | 20 (14.71%) | 3 (3.41%) | 15 (3.23%) | 22 (6.19%) |
| | 62 or more | 31 (2.97%) | 17 (12.50%) | 5 (5.68%) | 1 (0.22%) | 8 (2.24%) |
| **Sex** | Female | 777 (74.50%) | 108 (79.41%) | 64 (72.73%) | 338 (72.84%) | 268 (75.28%) |
| **Region** | Southwest | 743 (71.25%) | 113 (83.09%) | 66 (75.00%) | 301 (64.87%) | 263 (74.16%) |
| | South | 160 (15.34%) | 6 (4.41%) | 8 (9.09%) | 97 (20.91%) | 49 (13.76%) |
| | North | 16 (1.53%) | 3 (2.21%) | 5 (5.68%) | 3 (0.65%) | 5 (1.40%) |
| | Northeast | 79 (7.57%) | 10 (7.35%) | 8 (9.09%) | 36 (7.76%) | 25 (7.02%) |
| | Mid-West | 45 (4.31%) | 4 (2.94%) | 1 (1.14%) | 27 (5.82%) | 13 (3.65%) |
| **Education** | High school graduate or lower | 82 (7.86%) | 24 (17.65%) | 9 (10.23%) | 25 (5.39%) | 24 (6.78%) |
| | Incomplete Higher Education | 264 (25.31%) | 43 (31.62%) | 24 (27.27%) | 100 (21.55%) | 97 (27.32%) |
| | Bachelor's degree | 328 (31.45%) | 29 (21.32%) | 25 (28.41%) | 168 (36.21%) | 106 (29.86%) |
| | Postgraduate specialization | 195 (18.70%) | 23 (16.91%) | 16 (18.18%) | 92 (19.83%) | 64 (18.02%) |
| | Master's or Doctorate degree | 174 (16.68%) | 17 (12.50%) | 14 (15.91%) | 79 (17.03%) | 64 (18.02%) |
| **Monthly family income** | Less than 1 minimum wage | 26 (2.59%) | 3 (2.21%) | 2 (2.27%) | 10 (2.16%) | 11 (3.10%) |
| | 1 to less than 2 minimum wages | 101 (9.67%) | 21 (15.44%) | 7 (7.95%) | 41 (8.84%) | 32 (9.01%) |
| | 2 to less than 5 minimum wages | 311 (29.79%) | 49 (22.06%) | 25 (28.41%) | 138 (29.74%) | 99 (27.89%) |
| | 5 to less than 10 minimum wages | 337 (32.28% | 30 (22.06%) | 33 (37.50%) | 147 (31.68%) | 127 (35.77%) |
| | 10 to 20 minimum wages | 193 (18.49%) | 21 (15.44%) | 17 (19.32%) | 93 (20.04%) | 62 (17.47%) |
| | More than 20 minimum wages | 75 (7.18%) | 12 (8.82%) | 4 (4.55%) | 35 (7.54%) | 24 (6.76%) |

Of the total participants, 603 (57.81%) reported that they or someone in their household had experienced foodborne illness at least once in their lifetime. Results from the final model investigating risk factors for its occurrence is presented in Table 2. In the final model, respondents aged 62 years or older had lower odds to report foodborne illness compared to those aged 18–28 years (P = 0.043). Males also exhibited lower odds of reporting foodborne illness compared to females (P = 0.004). Respondents from the North and Mid-West regions of Brazil had significantly higher odds of foodborne illness compared to those from the Southeast region (P = 0.033 and P = 0.015, respectively). Finally, there was significantly higher odds of foodborne illness reports within members of Cluster 3, which was the cluster characterized by the poorest food handling practices, compared to members of Cluster 1 (OR = 1.62, P = 0,026). Likewise, Cluster 4 also demonstrated significantly higher odds of foodborne illness compared to Cluster 1 (OR = 1.83; P = 0.006). Although income was suspected to be a confounder in the causal diagram (Supplementary Material, S1 Fig), its exclusion did not result in significant coefficient changes, leading to its removal from the final model.

**Table 2. Results from final multivariable logistic regression model investigating risk factor for the occurrence of foodborne illness within survey respondents.**

| Variable | Category | OR | SE | 95% CI | P |
|---|---|---|---|---|---|
| Cluster 2 | | 1.09 | 0.31 | (0.62, 1.91) | 0.765 |
| Cluster 3 | | 1.62 | 0.35 | (1.06, 2.48) | 0.026 |
| Cluster 4 | | 1.83 | 0.40 | (1.19, 2.80) | 0.006 |
| Age | 29–39 | 1.15 | 0.20 | (0.81, 1.62) | 0.432 |
| | 40–50 | 1.19 | 0.25 | (0.78, 1.81) | 0.418 |
| | 51–61 | 0.99 | 0.30 | (0.55, 1.80) | 0.984 |
| | 62 or more | 0.41 | 0.18 | (0.17, 0.97) | 0.043 |
| Sex | Male | 0.65 | 0.10 | (0.49, 0.87) | 0.004 |
| Region | South | 0.91 | 0.16 | (0.64, 1.30) | 0.599 |
| | North | 4.06 | 2.67 | (1.12, 14.74) | 0.033 |
| | Nordeste | 1.37 | 0.34 | (0.83, 2.24) | 0.216 |
| | Mid-West | 2.41 | 0.87 | (1.19, 4.89) | 0.015 |
| Education | High school graduate or lower | 0.75 | 0.20 | (0.45, 1.26) | 0.283 |
| | Incomplete Higher Education | 1.53 | 0.30 | (1.05, 2.25) | 0.027 |
| | Postgraduate specialization | 1.04 | 0.20 | (0.71, 1.51) | 0.846 |
| | Master's or Doctorate degree | 1.19 | 0.24 | (0.81, 1.76) | 0.362 |

The reference category for the clusters was set as Cluster 1, as it exhibited the best adherence to food safety practices, making it a logical baseline for comparison. For other variables, the reference category was chosen based on the highest proportion of respondents.

## Discussion

This is the first large-scale study to use clustering analysis to evaluate the relationship between food handling practices, respondent profiles, and the likelihood of self-reported foodborne illnesses. While two previous studies have explored similar correlations between food manipulation behaviors and respondent characteristics with self-reported foodborne illness likelihood [10,25], these studies have fewer respondents (373 and 307 respectively) and did not use clustering analysis.

The study captured responses from individuals of diverse characteristics and environments. However, due to the academic setting where the survey was predominantly disseminated, the educational level of respondents was higher than what is typically observed in the general Brazilian population. The majority of respondents (71.25%) resided in the Southeast region of Brazil, which coincides with the location of Vila Velha University and the researcher's residence. Since the questionnaire was heavily distributed through social media, the majority of respondents (75.93%) were younger than 40 years old.

Foodborne illnesses represent a significant public health challenge in Brazil, with the number of reported cases likely underestimating the true incidence of foodborne infections [26]. Key contributing factors include inadequate hygiene practices, cross-contamination, and insufficient cooking, which collectively facilitate the transmission of pathogens through food consumption [27]. From 2014 to 2023, a total of 110,614 foodborne illness were recorded across Brazil, resulting in 121 fatalities [28]. This study revealed significant gaps in food handling practices across all respondents of our survey, highlighting multiple areas for potential intervention. Among the respondents of our survey, 677 (64.90%) reported using the same cutting board for both raw meat and vegetables and 964 (92.43%) indicated washing the cutting board before switching between different food types. Even though people probably mean well when performing this last practice, many individuals may not adequately clean their cutting boards [29]. Prolonged use of wooden and plastic cutting boards, which are the most commonly used in Brazil, often results in the formation of small fissures [30] that are difficult to

sanitize, providing an ideal environment for bacterial growth and biofilm formation. Consequently, this can facilitate cross-contamination between raw meat and fresh produce, posing a significant food safety risk [31].

Only 341 respondents (32.70%) reported washing packages before storing them in the refrigerator. Packaging from supermarkets or street food vendors is often heavily contaminated due to prolonged exposure to air, human contact, and transportation vehicles. Placing these unwashed packages directly into the refrigerator increases the risk of contamination and contributes to equipment soiling [32]. Additionally, 660 respondents (63.28%) indicated that they do not monitor their refrigerator's operating temperature. Refrigerators running at elevated temperatures may facilitate the growth of meso-philic bacteria, some of which are pathogenic [33]. Introducing external bacteria into the refrigerator and failing to ensure that it operates at a temperature low enough to inhibit bacterial proliferation can significantly increase the risk of foodborne illness within a household [34].

A total of 438 respondents (42.00%) reported consuming raw or soft-boiled eggs. Unlike in some countries where Salmonella-free eggs are widely available, such products are uncommon in Brazil, heightening the risk of salmonello-sis linked to undercooked eggs [35]. Proper cooking is critical to inactivate Salmonella spp., requiring both the egg yolk and white to reach and maintain a temperature of at least 74°C for several seconds [36]. Additionally, improper storage practices exacerbate this risk, as temperatures exceeding 8°C allow Salmonella spp. to proliferate within the egg yolk during storage [37]. In our study, 112 respondents (10.74%) reported storing their eggs outside the refrigerator, while 260 (24.93%) stored them on the refrigerator door. Studies indicate that the refrigerator door is typically the warmest area, often exceeding the temperature necessary to inhibit bacterial growth effectively [38,39]. This combination of consuming raw or undercooked eggs and inadequate cold chain management represents a substantial risk for foodborne illnesses.

A particularly striking issue was the widespread lack of thermometer use to verify meat doneness, a practice virtually absent among respondents. Most people, in the absence of a thermometer, rely solely on visual cues, such as the color of the meat's center, which is a poor indicator of safe cooking temperatures [40]. The consumption of raw or undercooked meat is a well-documented risk factor for foodborne illness [41]. Combined with the lack of thermometer use, this practice represents a substantial risk for foodborne illness. While the low utilization of thermometers in food preparation is con-cerning, their use alone cannot guarantee the safety of all consumed foods. Some toxins, such as cereulide, an emetic and thermotolerant toxin produced by Bacillus cereus, remain stable under cooking conditions [42]. Similarly, Clostridium perfringens can form heat-resistant spores that survive proper cooking and later germinate in improperly stored foods, producing toxins that affect the gastrointestinal tract [43].

Within Cluster 1, one notable issue among participants was that a significant proportion (58.09%) used trash bins on kitchen countertops. Kirchner et al. [44] highlighted that trash bin lids were among the most contaminated surfaces in kitch-ens and were significant contributors to cross-contamination. The proximity of trash bins to food preparation areas, coupled with frequent contact with raw food packaging and kitchen waste, makes them critical points of contamination. This finding underscores the need to address knowledge gaps, even among those who generally adhere to good food handling prac-tices. Another critical issue highlighted amongst participants from Clusters 1 and 2 was the practice of washing raw meat before cooking. Washing meat is not recommended, as it does not effectively remove pathogens and can spread bacteria up to 70 centimeters from the washing area [45]. Additionally, wet raw meat increases bacterial transfer to surfaces compared to dry meat [46]. The prevalence of this practice among respondents who otherwise followed good food handling practices suggests a widespread lack of awareness about the risks of washing raw meat. Food safety campaigns, such as those described by Henley et al. [47], could effectively target this specific behavior and reduce the associated risks.

All four clusters included respondents who consumed products beyond their secondary shelf life, but cluster 3 was the only group in which the majority (61.85% vs 33.09–48.86) engaged in this behavior. Consuming expired products, partic-ularly dairy and deli meats, poses significant risks. For instance, milk and sliced cooked ham past their secondary shelf life often harbor high levels of microbial contaminants, including members of Enterobacteriaceae family, which are major causes of foodborne illnesses [48,49].

Respondents aged 62 years or older had significantly lower odds of experiencing foodborne illness compared to those aged 18–28 years. While this result aligns with Brugeff et al. [14], other studies suggest that younger individuals tend to adopt better food handling practices [9,50–53].

Male respondents were less likely to report having experienced foodborne illness compared to female respondents, a finding that diverges from most previous research [8,10,14,17,53,54]. One possible explanation could be higher self-reporting tendency among females, which could influence our results [55]. Women may be more aware of or attentive to symptoms of foodborne illness due to their roles in food preparation and household health management, as suggested in prior studies on gender differences in health awareness [56–58]. Another contributing factor could be regional and cultural variability in food consumption and preparation practices. Given that most respondents concentrated in the Southeast region, known for its urbanized and structured food systems, gender-specific roles in food handling may have influenced exposure risks differently [59].

The North and Mid-West regions of Brazil exhibited a notable increase in the reported odds of foodborne illness. These regions face significant structural challenges that may contribute to higher incidences of foodborne illness [6]. Limited access to clean water and adequate sanitation remains a pressing issue, with over 33 million Brazilians lacking safe water and around 115 million living without improved sanitation [60]. Such deficiencies disproportionately impact rural and underserved areas, exacerbating the risk of food contamination. Moreover, these regions often have less developed food safety infrastructure and monitoring systems compared to the more urbanized Southeast and South regions. Local variability in food safety practices, coupled with limited awareness and adherence to safety guidelines, further amplifies the vulnerability to foodborne illnesses [61,62].

Our study showed elevated odds of foodborne illness in Cluster 4 compared to Cluster 1, despite similar levels of adherence to good practices. This outcome could be due to critical failures identified in questions captured in Block 5 of the survey, particularly concerning the consumption of raw and undercooked meat and eggs. Ducrocq et al. [63] demonstrated that individuals consuming undercooked meat had a higher likelihood of testing positive for Toxoplasma gondii antibodies compared to those consuming well-cooked meat. The consumption of undercooked meat is also associated with a higher seroprevalence of anti-hepatitis E virus antibodies [64]. Undercooked meat consumption is also linked to the transmission of Escherichia coli pathogenic strains responsible for gastrointestinal symptoms, such as E. coli O157:H7 and other Shiga toxin-producing E. coli (STEC). Cases typically result in abdominal cramps, vomiting, and/or diarrhea, which may progress to haemorrhagic colitis [65,66]. Additionally, the ingestion of undercooked poultry is one of the leading causes of Salmonella spp. and Campylobacter spp. contamination. Although salmonellosis is rarely fatal, it can result in severe gastrointestinal distress, hospitalization, and irritable bowel syndrome. It can also be life-threatening for immunocompromised individuals, such as those with HIV or cancer, as well as older adults. Similarly, infections caused by Campylobacter spp., though seldom fatal, place individuals at risk for other medical complications, including irritable bowel syndrome and Guillain-Barré Syndrome [67,68]. Undercooked meat consumption is also a significant risk factor for the acquisition of toxoplasmosis in humans [69]. These infections are often asymptomatic or cause mild, self-resolving symptoms such as fever, fatigue, and swollen lymph nodes. However, in individuals with weakened immune systems or pregnant women, the effects can be far worse [70].

Besides bacteria, viruses and fungi also are contribute to food poisoning worldwide [71]. Noroviruses is one major cause of foodborne illness with transmission often occurring through contaminated water used at various stages of food production or through food handlers who fail to adhere to proper hand hygiene practices [72]. Infections typically cause acute gastroenteritis [73]. Although less common than bacterial and viral pathogens, certain fungi, such as Aspergillus spp and Candida spp, can also lead to foodborne illnesses [74]. These infections are particularly concerning in immunosuppressed individuals, where they may primarily affect the gastrointestinal tract but can also manifest in more severe systemic symptoms [75].

Several similarities and differences emerge when comparing the findings of this study to those conducted in different cultural and geographical contexts. For instance, the prevalence of thermometer usage to check meat doneness was

similarly low in both this study (3.07%) and a study involving Serbian university students (8%) [18]. Additionally, the practice of using separate cutting boards for raw meat and vegetables was reported by 35.15% of respondents in this study, aligning with the 26.7% observed among Bangladeshi university students [76]. However, marked differences were noted in other food safety practices. While only 7.95% of respondents in this study washed eggs before storing them, over 40% of Polish consumers reported engaging in this practice [8]. Similarly, the habit of defrosting meat in the refrigerator was significantly higher in this study (56.99%) compared to studies conducted in Bangladesh (7%) [76], Lebanon (28%) [77], and Greece (25%) [78]. Handwashing before preparing food, a key hygiene practice, was reported by 45.88% of respondents in this study, contrasting with 60% in Slovenia [17], 32.6% in Ethiopia [79], and 55.3% in China [80]. Additionally, 45.11% of respondents in this study stored food arbitrarily within the refrigerator, a much lower proportion compared to 94% observed in Slovenia [17]. These comparisons highlight both the universal challenges in adopting food safety practices and the unique cultural and regional variations influencing behaviors.

To address the gaps identified in food handling practices within family residences, it is imperative to prioritize targeted interventions that extend beyond traditional public health campaigns. Future initiatives should integrate innovative educational tools, such as interactive digital platforms or community-based workshops, to increase awareness about critical practices like avoiding cross-contamination and ensuring proper food storage. Additionally, fostering collaboration between public health authorities and academic researchers could help design evidence-based programs tailored to regional specificities, addressing disparities such as limited access to resources in rural or underserved areas.

Future research should prioritize clinically or laboratory-confirmed foodborne illness cases to validate associations identified in self-reported data. Incorporating objective diagnostic criteria especially combined with prospective study designs (e.g., cohort studies) would reduce misclassification bias and enable more precise estimation of risks. It would also enable robust comparisons with the present data and investigation of the impact of interventions aimed at improving food handling practices, particularly within the most vulnerable groups identified in this study. Future studies should also consider probabilistic sampling methods and targeted recruitment strategies to capture more diverse regional, socioeconomic, and educational profiles. Assessing the long-term impact of these interventions and exploring the role of sociocultural factors in shaping household food safety behaviors is crucial. By building upon the insights from this study, a more comprehensive strategy can be developed to reduce foodborne illnesses at the household level, ensuring safer practices for future generations.

This study is not without limitations. Although it captured responses from individuals with diverse backgrounds, the use of a non-random sampling strategy and the predominance of survey dissemination within an academic setting likely contributed to a higher educational level among respondents compared to the general Brazilian population. Likewise, the majority of respondents (71.25%) resided in the Southeast region of Brazil, which coincides with the location of Vila Velha University and the researcher's geographical area of reach. This could reduce the generalizability of our findings, and readers should use caution when extrapolating results. Furthermore, it is important to note that the occurrence of foodborne illness was not confirmed by laboratories and relied on participant's recall of these events. This could lead to misclassification of the outcome, and the impact of this issue on analysis results is difficult to predict. Respondents may fail to accurately remember episodes, particularly if the symptoms were mild or occurred a long time ago. Additionally, respondents may have confused foodborne illness with other causes of gastrointestinal discomfort, such as food allergies or intolerances. Associations identified between food handling behaviors and self-reported foodborne illness should not be interpreted as strictly causal. This potential for misclassification of outcomes should be considered when interpreting the results.

## Conclusions

This study highlighted significant gaps in food handling practices within Brazilian households, suggesting the need for targeted educational interventions that may contribute to improved food safety and reduction of foodborne illness. The

 

cluster analysis revealed distinct behavioral patterns among respondents, with cluster 1 exhibiting the highest adherence to recommended practices, while cluster 3 demonstrated the lowest. Interestingly, despite comparable adherence to good practices, cluster 4 showed a higher likelihood of foodborne illness compared to cluster 1, likely largely due to risky behaviors such as the consumption of raw and undercooked foods. These findings emphasize the multifaceted nature of food safety, where adherence to hygiene and handling guidelines must be complemented by addressing high-risk behaviors to effectively reduce disease incidence.

The study highlights systemic issues across all clusters, including the widespread neglect of thermometer use to ensure proper meat cooking and the prevalent habit of consuming foods beyond their secondary shelf life. These behaviors significantly elevate the risk of exposure to pathogens. The insights gained from this research reinforce the critical role of food safety education in mitigating these risks. Public health campaigns tailored to address specific gaps identified in this study—such as discouraging meat washing, promoting thermometer usage, and providing clear guidance on food storage—can foster safer practices. Ultimately, these efforts have the potential to reduce the burden of foodborne illness in domestic settings and advance public health outcomes in Brazil.

## Supporting information

**S1 Table. Full questionnaire responses: Distribution of answer counts and percentages across clusters and total respondents.**
(DOCX)

**S1 Fig. Causal diagram depicting the relationship between respondents' profile and foodborne intoxication.** Causal diagram depicting the relationship between respondents' profile and foodborne illness. The arrows represent the direction and relative strength of influence, categorized as follows: red arrows indicate the weakest influence, yellow arrows represent moderate influence, and green arrows signify the strongest influence.
(TIF)

**S1 File. Excel copy of all answers to the questionnaire, with column names translated to English and any identifying data (name, e-mail) excluded.**
(XLSX)

## Acknowledgments

We would like to express our gratitude to all participants who took part in this study, as well as to those who assisted in disseminating the survey through social media platforms and other channels.

## Author contributions

**Conceptualization:** Gustavo Guimarães Fernandes Viana, Gabriel Augusto Marques Rossi.

**Data curation:** Gustavo Guimarães Fernandes Viana, Andréia Gonçalves Arruda, Gabriel Augusto Marques Rossi.

**Formal analysis:** Gustavo Guimarães Fernandes Viana, Andréia Gonçalves Arruda, Gabriel Augusto Marques Rossi.

**Funding acquisition:** Gabriel Augusto Marques Rossi.

**Investigation:** Gustavo Guimarães Fernandes Viana, Andréia Gonçalves Arruda, Gabriel Augusto Marques Rossi.

**Methodology:** Gustavo Guimarães Fernandes Viana, Andréia Gonçalves Arruda, Gabriel Augusto Marques Rossi.

**Project administration:** Gabriel Augusto Marques Rossi.

**Supervision:** Gabriel Augusto Marques Rossi.

**Writing – original draft:** Gustavo Guimarães Fernandes Viana, Andréia Gonçalves Arruda, Gabriel Augusto Marques Rossi.

**Writing – review & editing:** Gustavo Guimarães Fernandes Viana, Andréia Gonçalves Arruda, Gabriel Augusto Marques Rossi.

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
