## [Decision Letter · Decision Letter 0]

Dear Dr. Arruda,

Thank you for submitting your manuscript to PLOS ONE. After careful consideration, we feel that it has merit but does not fully meet PLOS ONE’s publication criteria as it currently stands. Therefore, we invite you to submit a revised version of the manuscript that addresses the points raised during the review process.

We look forward to receiving your revised manuscript.

Kind regards,

António Raposo

Academic Editor

PLOS ONE

“Fundação de Amparo à Pesquisa e Inovação do Espírito Santo (FAPES) (Grant 551/2023 P 2023-RH7P2)”

Reviewers' comments:

Reviewer's Responses to Questions

**Comments to the Author**

1. Is the manuscript technically sound, and do the data support the conclusions?

Reviewer #1: Yes

Reviewer #2: Yes

Reviewer #3: Yes

2. Has the statistical analysis been performed appropriately and rigorously?

Reviewer #1: Yes

Reviewer #2: Yes

Reviewer #3: Yes

3. Have the authors made all data underlying the findings in their manuscript fully available?

Reviewer #1: No

Reviewer #2: Yes

Reviewer #3: Yes

4. Is the manuscript presented in an intelligible fashion and written in standard English?

Reviewer #1: Yes

Reviewer #2: Yes

Reviewer #3: Yes

Reviewer #1: The manuscript presents a technically sound study investigating behavioral patterns related to food safety and their association with self-reported foodborne illness. The study is well-structured, employs rigorous statistical methods, and provides valuable insights into food safety behaviors. However, certain limitations, particularly in data availability and generalizability, should be addressed.

The study's methodology is appropriate, and statistical analyses have been conducted rigorously. The use of logistic regression, Spearman correlation, and cluster analysis is suitable for identifying behavioral patterns and risk factors. Ethical approval has been obtained, ensuring adherence to research guidelines. However, the reliance on self-reported foodborne illness cases introduces potential recall bias and misclassification risks. Additionally, the study sample is not fully representative, which may affect the broader applicability of findings.

Acknowledge the potential impact of recall bias more explicitly in the discussion.

If possible, provide additional justification for the representativeness of the sample or discuss how future studies could address this limitation.

The conclusions align with the study's results, emphasizing the need for food safety education and intervention programs. However, due to the reliance on self-reported data, a causal relationship between behaviors and foodborne illness cannot be definitively established. The authors correctly acknowledge this limitation, but it should be further emphasized in the discussion.

Consider softening causal language in the conclusion to reflect the observational nature of the study.

Highlight future research directions, such as using objective, clinically/laboratory-confirmed cases to validate findings.

The statistical analysis has been performed rigorously and appropriately. The chosen methods are well-suited for the study objectives and provide meaningful insights into food safety behaviors. No major concerns were found in the statistical approach.

Recommendations:

Provide a publicly accessible dataset containing anonymized individual data points or summary statistics.

If restrictions exist due to privacy concerns, clearly justify the limitations and provide alternative access options, such as upon reasonable request.

The manuscript is written in clear and standard English. The text is generally intelligible, with no major grammatical or typographical errors noted.

The manuscript is technically sound and presents meaningful research findings. However, data availability issues need to be addressed to meet journal requirements. Additionally, some minor revisions in the discussion and conclusion would strengthen the manuscript.

Reviewer #2: This study provides valuable insight into food handling behaviors in Brazilian households and their potential link to self-reported foodborne illness.

The cross-sectional nature of the study limits its ability to establish causality. While it identifies associations between behaviors and self-reported illness, it cannot definitively claim that poor food handling caused these illnesses

Reviewer #3: This study presents a significant advancement in understanding food safety and foodborne illness risk factors by specifically examining household practices in Brazil, an area that has been less explored compared to commercial settings, which dominate the existing literature.

The manuscript appears technically sound, with data supporting the conclusions drawn regarding food safety practices and the incidence of foodborne illnesses among Brazilian households. The statistical analysis is conducted rigorously, employing appropriate methods such as cluster analysis and logistic regression to investigate the relationships between various factors and the self-reported occurrence of foodborne illnesses. While the authors indicate that data underlining the findings will be made available and provide a general availability statement, they do not specify the means by which this data can be accessed, which may limit transparency. The manuscript is presented in an intelligible manner, adhering to academic writing standards with clear articulation of methodologies and findings in standard English. Overall, it effectively communicates its objectives and results to the intended audience.

Note: Please attend to the error in the range given on page 5, line 186.

**Do you want your identity to be public for this peer review?** For information about this choice, including consent withdrawal, please see our Privacy Policy

Reviewer #1: **Yes: ** Hala Awad

Reviewer #2: **Yes: ** M. João Lima

Reviewer #3: No

---

## [Author Response · Author response to Decision Letter 1]

24 Apr 2025

This has been provided in a separate document.

---

## [Decision Letter · Decision Letter 1]

Assessing Food Safety Practices and Foodborne Illness Risk Factors in Brazilian Households

PONE-D-25-15304R1

Dear Dr. Arruda,

We’re pleased to inform you that your manuscript has been judged scientifically suitable for publication and will be formally accepted for publication once it meets all outstanding technical requirements.

Kind regards,

António Raposo

Academic Editor

PLOS ONE

Additional Editor Comments (optional):

Reviewers' comments:

Reviewer's Responses to Questions

**Comments to the Author**

Reviewer #2: All comments have been addressed

2. Is the manuscript technically sound, and do the data support the conclusions?

Reviewer #2: Yes

3. Has the statistical analysis been performed appropriately and rigorously?

Reviewer #2: Yes

4. Have the authors made all data underlying the findings in their manuscript fully available?

Reviewer #2: Yes

5. Is the manuscript presented in an intelligible fashion and written in standard English?

Reviewer #2: Yes

Reviewer #2: This version of the article shows a significant improvement, as evidenced by more fluid writing, with some less dense and more explicit details. It is therefore publishable.

**Do you want your identity to be public for this peer review?** For information about this choice, including consent withdrawal, please see our Privacy Policy

Reviewer #2: **Yes: ** M. João Reis Lima

---

## [Editor Report · Acceptance letter]

PONE-D-25-15304R1

PLOS ONE

Dear Dr. Arruda,

I'm pleased to inform you that your manuscript has been deemed suitable for publication in PLOS ONE. Congratulations! Your manuscript is now being handed over to our production team.

Kind regards,

on behalf of

Dr. António Raposo

Academic Editor

PLOS ONE